# Neurocognitive Impulsivity in Opiate Users at Different Lengths of Abstinence

**DOI:** 10.3390/ijerph20021236

**Published:** 2023-01-10

**Authors:** Elena Psederska, Jasmin Vassileva

**Affiliations:** 1Bulgarian Addictions Institute, 93 Antim I Str., 1303 Sofia, Bulgaria; 2Department of Cognitive Science and Psychology, New Bulgarian University, 1618 Sofia, Bulgaria; 3Institute for Drug and Alcohol Studies, Virginia Commonwealth University, Richmond, VA 23219, USA; 4Department of Psychiatry, Virginia Commonwealth University, Richmond, VA 23219, USA

**Keywords:** decision making, response inhibition, opiate use, length of abstinence

## Abstract

The aim of the current study was to examine the effects of length of abstinence on decision making (impulsive choice) and response inhibition (impulsive action) in former opiate users (OU). Participants included 45 OU in early remission [0–12 months of abstinence], 68 OU in sustained remission [>12 months of abstinence], and 68 control participants. Decision making was assessed with the Iowa Gambling Task (IGT), the Cambridge Gambling Task (CGT), and the Monetary Choice Questionnaire (MCQ). Response inhibition was examined with the Stop Signal Task (SST), and the Go/No-Go Task (GNG). Results revealed group differences in decision making under risk (CGT) and ambiguity (IGT), where control participants displayed better decision making compared to OU in early remission. Both groups of former OU were also characterized by higher discounting of delayed rewards (MCQ). Regression analyses revealed minimal effects of length of abstinence on performance on decision-making tasks and no effects on delay discounting. In addition, both OU groups showed reduced action inhibition (GNG) relative to controls and there were no group differences in action cancellation (SST). Length of abstinence had no effect on response inhibition. Overall, our findings suggest that neurocognitive function may not fully recover even with protracted abstinence, which should be addressed by relapse prevention and cognitive remediation programs for OU.

## 1. Introduction

### 1.1. Neurocognitive Impulsivity and Substance Use Disorders

Substance use disorders (SUDs) are recognized as chronic relapsing brain disorders, associated with specific neuroadaptive changes in brain functioning and corresponding long-lasting neurocognitive impairments [1,2,3]. Impulsivity and inhibitory control are among the neurofunctional domains most severely affected by chronic substance use [2]. Impairments in inhibitory and impulse control are widely accepted as both antecedent vulnerabilities contributing to the development and maintenance of SUDs, as well as negative outcomes resulting from chronic drug use, associated with persistent neuroadaptive changes in the brain that may affect the ability to maintain long-term abstinence [4,5,6].

Neurocognitive impulsivity is a multidimensional domain reflecting state-dependent cognitive and emotional processes associated with reward processing and behavioral inhibition [7,8], typically measured by performance-based laboratory tasks. The neurocognitive domain of impulsivity is further divided into two main subdomains reflecting distinct neuroanatomical and neuropharmacological mechanisms [8,9,10,11,12].

*Impulsive action* is the first subdomain of neurocognitive impulsivity, associated with deficits in rapid response inhibition [13]. It is usually measured by Stop Signal Tasks (SST) [14,15] involving cancellation of an already initiated motor response (i.e., action cancellation), and/or Go/No-Go paradigms (GNG) [16], requiring the inhibition of a dominant behavioral reaction (i.e., action inhibition) [10]. Although the majority of studies in the literature have used Stop Signal and Go/No-Go paradigms as interchangeable alternatives measuring the same latent process (i.e., response inhibition), a growing number of researchers have emphasized that these tasks engage heterogeneous cognitive processes mediated by distinct neural circuits, which requires their separate evaluation in future studies [17,18,19].

*Impulsive choice* is the second subdomain of neurocognitive impulsivity, related to reward-driven and impulsive decision making, increased risk-taking, and preference for smaller immediate over larger delayed rewards [8,20]. Common measures of impulsive choice include delay discounting tasks such as the Monetary Choice Questionnaire (MCQ) [21] and computerized gambling tasks such as the Iowa Gambling Task (IGT) [22], examining decision making under ambiguity, and the Cambridge Gambling Task (CGT) [23], capturing decision making under risk. Accumulating evidence suggests that although impulsive choice tasks measure temporally stable decision-making ability, different tasks capture different subprocesses regulated by distinct neural circuits [24,25].

Deficits in impulsive choice and impulsive action are among the most notable and common neurocognitive impairments in substance dependent individuals (SDIs). A plethora of studies indicate that SDIs are characterized by deficient inhibitory control (for review see [26]), compromised decision making, and steeper delay discounting [27,28,29]. In addition, neurocognitive deficits in impulsivity have been consistently associated with negative treatment outcomes, suggesting that they may hinder the ability to successfully achieve and maintain long-term abstinence [30,31,32,33,34].

### 1.2. Neurocognitive Impulsivity in Opiate Users

Although SDIs demonstrate notable impairments on most tasks of impulsive choice and impulsive action, recent studies suggest that neurocognitive deficits may be further influenced by the unique properties of different classes of psychoactive substances. Consistent with the precision medicine approach, current addiction models emphasize the growing need to identify substance-specific neurocognitive profiles, which may further increase the precision of addiction prevention and intervention programs [8,35,36].

Despite the recently increased focus on elucidating the specific effects of different classes of drugs on neurocognitive function, the role of impulsivity has been more thoroughly investigated in stimulant- and alcohol use disorder [37,38], whereas our knowledge about its role in opiate use disorder (OUD) remains inconclusive. This reflects a global trend in the addiction literature associated with greater research attention to other SUDs relative to opiates [8]. Although chronic stimulant use is associated with more pronounced neuroadaptations in dopaminergic pathways and more severe neurocognitive impairments [26,39], deficits in inhibitory control and decision making appear to be an integral feature of opiate addiction as well [40].

Studies examining impulsive action among opiate dependent individuals have been highly inconsistent. Some studies have found impaired response inhibition among opiate users [41,42,43], whereas others have failed to find any performance differences between opiate users and control participants [44,45,46]. In addition, other SUDs appear to be more strongly associated with response inhibition deficits compared to OUD [46,47]. These findings suggest that deficits in response inhibition are not a key discriminating feature of opiate addiction, which has been further supported by machine learning studies revealing that opiate dependence is uniquely predicted by relatively intact response inhibition [48].

Studies on impulsive choice in opiate users have found notable impairments in decision making [41,49,50,51,52,53,54] and delay discounting [21,50,55]. In addition, neuroimaging studies with opiate dependent individuals have shown that deficits in impulsive choice are associated with long-lasting impairments in brain functioning that persist into protracted abstinence [56]. However, findings from studies comparing the performance of individuals dependent on different classes of drugs on impulsive choice tasks are somewhat conflicting. Some studies have reported that stimulant users are characterized by compromised decision making compared to opiate users [23,47,55,57], whereas others have found no performance differences on impulsive choice tasks between opiate and stimulant users [46,58,59]. Despite these contradictory findings, meta-analytic studies reveal that decision making is one of the most severely affected neurocognitive functions in opiate users [40]. In addition, some studies have found that risky decision making and increased delay discounting negatively affect the ability to successfully achieve and maintain long-term abstinence [31,60]. The predictive utility of distinct dimensions of neurocognitive impulsivity for clinically relevant outcomes (e.g., relapse, treatment adherence, treatment prognosis) requires their further investigation within the framework of opiate addiction. The accumulation of a rich evidence base may lead to the identification of opiate-specific profiles of neurocognitive impairments and the development of individualized treatment strategies that target specific neurocognitive functions affected by chronic opiate use, while enhancing individual resources related to areas of optimal neurocognitive functioning.

### 1.3. Neurocognitive Impulsivity and the Protracted Abstinence Stage of Addiction

One of the most understudied topics in the addiction literature is the relative stability or reversibility of neurocognitive impairments among former substance users. The question of whether cognitive deficits persist in protracted abstinence from chronic opiate use has been rarely addressed in research, which limits our understanding of the clinically relevant correlates of protracted abstinence and recovery of function.

The limited number of studies investigating neurocognitive impulsivity dimensions in abstinent opiate users have uncovered persistent impairments in inhibitory control and decision making. Previous studies examining impulsive choice among abstinent opiate users report impaired decision making and increased delay discounting [49,50,53,58,61,62]. Meta-analytical studies have concluded that length of abstinence has no effect on decision-making deficits in opiate dependent individuals, suggesting that recovery in these functions is minimal at best [63]. These findings are consistent with neuroimaging data showing chronic functional changes in prefrontal regions and long-lasting impairments in dopamine transmission in former opiate users, who have maintained successful abstinence for an average of 18 months (for review see [63]).

Data from studies evaluating the potential recovery of function of impulsive action among former opiate users have produced somewhat conflicting findings. Studies conducted with participants in the earlier stages of abstinence [between 2 and 8 months] have found impairments in response inhibition [43,64,65], whereas studies in the later stages of abstinence [>12 months] have reported relatively intact response inhibition capacity [45,48,66].

Previous studies examining impulsive choice and impulsive action in the protracted abstinence stage of opiate addiction are marked by several methodological limitations. First, the majority of studies were based on samples of active opiate users or individuals in the earlier stages of abstinence [<12 months], which impedes the identification of impairments in the later stages of protracted abstinence. Therefore, one potential explanation for the conflicting results in the literature may be related to the variability in length of abstinence across studies. Overall, the earlier stages of abstinence seem to be consistently associated with increased response disinhibition [43,45,65], whereas response inhibition appears to be intact in the latter stages of protracted abstinence [48,66]. On the other hand, deficits in impulsive choice seem to be robust in both earlier [49,61] and later stages of protracted abstinence [50,67]. Second, the vast majority of studies used samples consisting of individuals who met criteria for more than one SUDs or were in opioid maintenance treatment. These limitations prevent understanding the residual effects of opiate use on neurocognitive dimensions of impulsivity in protracted abstinence. A limited number of studies conducted with mono-dependent opiate users report intact response inhibition [45,48,68] and pervasive impairments in impulsive choice [49,50,53,69].

Future research focusing on the role of neurocognitive impulsivity in the protracted abstinence stage hold promise for uncovering clinically relevant patterns that may have real-world practical implications for addiction treatment and rehabilitation, as well as for the development of novel, personalized therapeutic interventions that may be used for treatment of opiate addiction. Identifying specific impairments in neurocognitive functioning among individuals who successfully maintain protracted abstinence may be key for the development of individualized rehabilitation programs aiming to strengthen and improve cognitive function, which might be critical in the context of the current opioid crisis.

Considering the identified limitations of prior research, the current study focused on examining the effects of length of abstinence on various neurocognitive dimensions of impulsivity among opiate monodependent individuals at different stages of abstinence—early full remission [<12 months] and sustained full remission [>12 months].

## 2. Materials and Methods

### 2.1. Participants

Data were collected as part of a larger international research project conducted in Bulgaria between 2010 and 2020 investigating different types of impulsivity in opiate and stimulant users. Participants were recruited via information handouts placed at various substance abuse treatment facilities and social venues in Bulgaria, as well as through personal referral and information about the study published on the study’s web page and Facebook page. All interested participants were screened via telephone on several medical and substance use eligibility criteria. All participants had to meet the following inclusion criteria to be enrolled in the study: (1) age between 18 and 50 years, (2) estimated IQ higher than 75 [70]; (3) minimum of 8th grade education; (4) being able to read and write in Bulgarian; (5) HIV-seronegative status; (6) negative breathalyzer test for alcohol and negative urine toxicology screen for amphetamines, methamphetamines, cocaine, opiates, methadone, cannabis, benzodiazepines, barbiturates, and MDMA. Exclusion criteria included history of neurological illness, head injury with loss of consciousness of more than 30 min, history of psychiatric disorders (i.e., psychotic disorders or mood disorders), current treatment with opioid agonists and/or use of medications that affect impulsivity (i.e., neuroleptics, antidepressants, benzodiazepines).

The sample consisted of 181 participants between the ages of 21 and 48 years. 113 participants met DSM-IV criteria for lifetime heroin mono-dependence, of whom 45 (39.8%) were in early full remission [<12 months of abstinence] and 68 (60.2%) were in sustained full remission [>12 months of abstinence] according to DSM-IV criteria [71]. The group of participants in early full remission included 9 (20%) females and 36 (80%) males. The group of participants in sustained full remission included 13 (19.1%) females and 55 (80.9%) males. None of the participants who met criteria for lifetime heroin mono-dependence were on opioid maintenance therapy at the time of testing. A total of 37 (32.7%) participants were enrolled in nonpharmacological treatments at the time of testing, which included mainly psychosocial interventions. The control group included 17 (25%) females and 51 (75%) males who had no history of substance abuse or dependence.

### 2.2. Procedures

The study was approved by the Institutional Review Boards of Virginia Commonwealth University and the Medical University in Sofia on behalf of the Bulgarian Addictions Institute. Subjects who met inclusion criteria based on the telephone screening interview were contacted and invited to take part in the study. Written informed consent was obtained from all participants enrolled in the study. Abstinence from alcohol and drug use at the time of testing was verified by breathalyzer test (Alcoscan AL7000) and urine toxicology screen for amphetamines, barbiturates, benzodiazepines, cannabis, cocaine, MDMA, methadone, methamphetamines, and opiates. All participants were HIV-seronegative, determined by rapid HIV testing.

Data were collected in two study sessions conducted on two separate days. The completion of each session takes approximately 4 h. The study protocol included a variety of clinical interviews, self-report scales and computer-based neurobehavioral tests for examining SUDs, neurocognitive functions, externalizing and internalizing personality traits and disorders. Detailed description of the study protocol and prior findings with the research project can be found in our previous publications [48,58,68,69,72]. Participants were paid a total of 80 Bulgarian leva (approximately 50 USD) for participation in the study.

### 2.3. Measures

#### 2.3.1. Assessment of SUDs

SUDs were examined using the *Structured Clinical Interview for DSM-IV-Substance Abuse Module* (SCID-SAM) [73]. The SCID-SAM is a clinical interview used to assess whether participants meet criteria for any SUD (alcohol-, cannabis-, stimulant-, hallucinogen-, opioid use disorders) listed in the DSM-IV [71]. The DSM-IV includes two related diagnostic categories reflecting different levels of severity of substance use problems: substance abuse and substance dependence. For a diagnosis of substance abuse, at least 1 of 4 symptoms must be met, whereas substance dependence requires the presence of at least 3 out of 7 symptoms. The diagnostic categories substance abuse and substance dependence have undergone major revisions in the DSM-5 [72] as they were merged into a single continuum representing Substance Use Disorders (SUD), with diagnosis no longer involving two dichotomous alternatives but being made dimensionally according to the severity of the presented symptoms. Given that data collection for the study started in 2010 prior to the publication of the DSM-5 in 2013, we used the DSM-IV criteria to evaluate the presence of opiate dependence symptoms. Trained raters coded the presence of DSM-IV symptoms of substance abuse and dependence on a three-point scale (0 = not present, 1 = subthreshold, 2 = present). Substance dependence was diagnosed if the participant met three (or more) of the seven criteria within a 12-month period. All opiate dependent participants in the current sample met criteria for lifetime opiate dependence but had no history of dependence on other classes of drugs. In addition, all opiate users were in protracted abstinence at the time of testing. Remission was specified according to the DSM-IV criteria: (1) *early full remission* was coded if the participant met none of the opiate abuse/dependence criteria for at least 1 month, but less than 12 months; (2) *sustained full remission* was coded if the participant met none of the opiate abuse/dependence criteria for at least 12 months; (3) *early partial remission* was coded if the participant did not meet full criteria for opiate dependence for at least 1 month, but less than 12 months, while some opiate abuse/dependence symptoms were present during this period; (4) *sustained partial remission* was coded if the participant did not meet full criteria for opiate dependence for at least 12 months, but some opiate abuse/dependence symptoms were present during this period. All opiate dependent participants in the current sample met criteria for full remission (early or sustained) and were divided in two groups accordingly. We used three OUD outcomes in the analyses: (1) severity of opiate dependence (i.e., number of OUD symptoms met); (2) duration of opiate dependence (in days); and (3) length of abstinence (in days).

#### 2.3.2. Assessment of Intelligence

The Raven’s Standard Progressive Matrices (RSPM) [70] were used to measure fluid intelligence. The RSPM include 60 multiple-choice visual stimuli, divided into 5 sections of 12 items each, ordered in increasing levels of difficulty. Test stimuli presented various patterns of shapes with a missing piece. Participants are instructed to identify the missing piece that completes each shape by selecting one out of six or eight options. IQ scores ranged from 55 to 132 points.

#### 2.3.3. Neurocognitive Measures of Impulsivity

##### Measures of Impulsive Choice

The *impulsive choice* domain was measured by three neurocognitive tasks that capture its distinct components: (1) decision making under ambiguity; (2) decision making under risk; and (3) delay discounting.

*The Iowa Gambling Task* (IGT) [22,74] is a computerized neurocognitive task that examines decision making under ambiguity and involves learning by trial-and-error. Four decks of cards are presented on the computer screen and participants are instructed to select cards from the decks to increase their earnings. Selecting from Decks A and B provides higher rewards but also higher occasional penalties. In contrast, selecting from Decks C and D is related to both lower earnings and penalties, representing a more advantageous decision-making strategy in the long-term. The IGT includes 100 trials or five blocks of 20 choices each. It has been suggested that performance on the task can be subdivided into two main phases measuring different decision-making processes: (1) an early “learning” phase, which covers the first two blocks of the task and measures decision making under ambiguity, and (2) a latter “risky” phase, which covers performance on the last three blocks of the task, and assesses decision making under risk [75,76]. The performance indices used in the current study were: (1) *IGT Net score*, reflecting the total number of advantageous choices minus the total number of disadvantageous choices over 100 trials (values range from −40 to 60 points); and (2) the combined net score of the last three blocks of the task (*IGT Blocks 3–5*; Trials 41–100), to control for the effects of learning on task performance.

*The Cambridge Gambling Task* (CGT) [23] is a computerized neurocognitive task that measures decision making under risk and does not require learning. In this task, 10 boxes colored red or blue are presented on the computer screen and participants are instructed to guess whether a yellow token is hidden under a red or a blue box to earn points. The ratios of red:blue boxes vary from 1:9 to 9:1 in pseudorandom order. In a second phase of the task participants bet from 5 to 95% of their earned points in ascending or descending order on their choices of red or blue boxes. For the purposes of the current study, we used the *CGT Quality of decision-making* index, which reflects betting on the more likely outcome of the two possible alternatives (i.e., red or blue boxes), as assessed by the percentage of instances in which the participant bet on the color that has the higher box ratio.

*The Monetary Choice Questionnaire* (MCQ) [21] is a self-report instrument that examines delay discounting. The MCQ is a 27-item forced choice measure in which participants are instructed to choose between smaller immediate rewards available on the day of testing and larger delayed rewards available from 1 week to 6 months in the future. The 27 items can be divided in three categories based on the approximate magnitudes of the delayed rewards: small ($25–35), medium ($50–60) and large ($75–85). The *MCQ Overall k* index was used to reflect the delay discounting rate (values range from 0.0003 to 0.2500). We used the log transformed value due to the non-normal distribution of MCQ scores in the current sample.

##### Measures of Impulsive Action

*Impulsive action* was assessed with two neurocognitive tasks that capture different components of response inhibition: (1) action inhibition (i.e., automatic inhibition) and (2) action cancellation (i.e., controlled inhibition).

*The Go/No-Go Task* (GNG) [16] is a computerized measure of response inhibition in conditions of varying difficulty (i.e., *automatic inhibition*). A series of two-element visual stimuli arrays are presented on the computer screen for 500 ms and participants are instructed to respond by pressing a button when the two elements are identical (“Go”) and to inhibit responding when the stimuli are discrepant (“No-Go”). The task includes 240 trials, 180 “Go”-conditions and 60 “No-Go”-conditions, which are further divided into two subconditions with varying levels of difficulty: in the “easy” No-Go condition (30 trials) the two visual stimuli are completely different, while in the “hard” No-Go condition (30 trials) the stimuli are very similar, with one stimulus mirroring the other. For the purposes of the current study, we used the general *GNG False Alarms* index, measuring incorrect responding to a non-target stimulus as the main index of response disinhibition. To analyze performance on the GNG task in more detail, we also used the separate indices of impulsive action based on the difficulty of the No-Go condition: (1) *GNG Easy No-Go False Alarms* and (2) *GNG Hard No-Go False Alarms*.

*The Stop Signal Task* (SST) [14,77] is a stop-signal paradigm examining action cancellation (i.e., *controlled inhibition*). In the task participants are presented with a series of five-digit numbers displayed on the computer screen for 500 ms each. Subjects are instructed to press a button when the stimulus currently presented on the screen matches the previous one (“Go”) and to inhibit responding when the stimulus is identical to the preceding one, but then changes color from black to red (“Stop”). Stop signals occur at 50, 150, 250, and 350 ms intervals following the appearance of the target stimulus. The performance measure used in the analyses was the *SST 150 ms Inhibition* ratio, calculated by dividing the failures to inhibit a response on “Stop trials” by correct detections on “Go trials” at the 150 ms stop-signal delay.

### 2.4. Data Analytic Plan

Our main aims were (1) to examine differences in impulsive choice and impulsive action among former opiate users at different lengths of abstinence and control participants with no history of substance dependence, and (2) to assess the effects of length of abstinence on different neurocognitive domains of impulsivity.

First, descriptive statistics and group differences in demographic characteristics and substance use variables are presented, followed by descriptive statistics and group differences in indices of impulsive choice and impulsive action. Group differences in demographics (i.e., age and years of education) and indices of decision making under risk (*CGT Quality of decision making*), delay discounting (*MCQ Overall k*) and impulsive action (*GNG False alarms*, *SST 150 ms Inhibition*) were assessed using a nonparametric Kruskal–Wallis test, due to the non-normal distribution of these variables in our sample. Group differences in fluid intelligence (i.e., Raven’s estimated IQ) and decision making under ambiguity (*IGT Net score*) were examined with One-Way ANOVA. Gender differences were assessed with chi-square analysis. Group differences in opiate use variables were examined with nonparametric Mann–Whitney U tests. Finally, a series of hierarchical multiple regressions were conducted to assess the effects of length of abstinence on distinct neurocognitive domains of impulsivity (i.e., impulsive choice and impulsive action). All regression analyses controlled for the effects of relevant covariates and followed the same steps: Step 1 included the demographic variables: biological sex, age, and fluid intelligence (Raven’s estimated IQ). Step 2 added the variables relevant to opiate use: duration of opiate dependence (in days) and severity of opiate dependence (number of DSM-IV opiate dependence symptoms met). Step 3 added length of abstinence (in days). All analyses were conducted in SPSS v26.0 using an alpha of 0.05.

## 3. Results

### 3.1. Descriptive Statistics and Group Differences in Demographic and Substance Use Variables

Participants included 113 former opiate users and 68 control participants with no history of substance use. Former opiate users were divided in two groups depending on their length of abstinence according to the DSM-IV remission criteria. The group of participants in early full remission [<12 months of abstinence] included 48 former opiate users who maintained abstinence for a mean of 6.56 (SD = ±3.76) months (range 30 days to 365 days). The group of participants in sustained full remission [>12 months of abstinence] included 68 former opiate users who maintained abstinence for a mean of 4.40 (SD = ±2.52) years (395 days to 3285 days (9 years)). Groups were well matched on most demographic and substance use variables. There were no significant differences in age [H_(2)_ = 3.68, *p* = 0.159] and gender [χ^2^_(2,N=181)_ = 0.78, *p* = 0.677] across groups; however, groups differed significantly in years of education [H_(2)_ = 29.18, *p* < 0.0001, ε^2^ = 0.162] and fluid intelligence [F_(2,177)_ = 8.73, *p* < 0.0001; η_p_^2^ = 0.090]. Dunn’s test with Bonferroni correction for multiple comparisons was used to examine specific group differences in years of education. Control participants reported higher education than both opiate dependent groups (*p*’s < 0.001). Scheffe’s post-hoc comparisons were used to assess group differences in fluid intelligence. Control participants had higher estimated IQ than both groups of heroin users (*p*’s < 0.01). Analyses examining differences in opiate use characteristics (i.e., duration of opiate dependence, severity of opiate dependence, and length of abstinence) between opiate users in early- [<12 months of abstinence] and sustained remission [>12 months of abstinence] revealed, as expected, that there were significant group differences only in length of abstinence [U = 3060.00, z = 8.981, *p* < 0.0001; η^2^ = 0.720]. We did not observe significant differences between the two opiate groups in severity of opiate dependence [U = 1306.50, z = −1.400, *p* = 0.161] and duration of opiate dependence [U = 1245.50, z = −1.674, *p* = 0.094]. Descriptive statistics and group differences in demographic and opiate use variables are presented in Table 1.

### 3.2. Descriptive Statistics and Group Differences in Impulsive Choice and Impulsive Action

#### 3.2.1. Impulsive Choice

Analyses testing for group differences in *impulsive choice* tasks revealed that the three study groups differed significantly on all *impulsive choice* indices: (1) *decision making under risk* (*CGT Quality of decision making* [H_(2)_ = 10.45, *p* = 0.005, ε^2^ = 0.059]), (2) *decision making under ambiguity* (*IGT Net score* [F_(2,172)_ = 4.43, *p* = 0.013, η_p_^2^ = 0.049]), and (3) *delay discounting* (*MCQ Overall k* [H_(2)_ = 14.30, *p* = 0.001, ε^2^ = 0.079]). Additional post-hoc analyses for pairwise comparisons showed that the control group was characterized by better decision making in both risky (i.e., CGT) and ambiguous (i.e., IGT) context compared to opiate users in early remission (*p*’s < 0.01). On the other hand, both groups of opiate dependent participants were characterized by increased delay discounting (i.e., MCQ) relative to the control group (*p*’s < 0.05). Table 2 provides descriptive statistics and group differences on indices of impulsive choice.

#### 3.2.2. Impulsive Action

Analyses assessing group differences in distinct domains of response inhibition revealed that the three study groups differed significantly in their ability to successfully inhibit a dominant behavioral response on tasks measuring *action inhibition* or *automatic inhibition* (*GNG False alarms* [H_(2)_ = 8.36, *p* = 0.015; ε^2^ = 0.048]). Post-hoc analyses showed that the control group was characterized by better automatic response inhibition compared to the two groups of opiate users (*p*’s < 0.05). Additional analyses assessing group differences in the quality of response inhibition in the separate GNG conditions (i.e., Easy No-Go and Hard No-Go) revealed that groups differed significantly only on the hard No-Go trials [H_(2)_ = 7.37, *p* = 0.025; ε^2^ = 0.042], where control participants performed better than the two opiate dependent groups (*p*’s < 0.05). No group differences were observed on the easy No-Go condition [H_(2)_ = 4.64, *p* = 0.098]. In addition, no differences were found between the three study groups in their ability to cancel an already initiated motor response on tasks measuring *action cancellation* or *controlled inhibition* (*SST 150 ms Inhibition* [H_(2)_ = 0.047, *p* = 0.792]). Table 3 provides descriptive statistics and group differences on indices of *impulsive action*.

### 3.3. Regression Analyses

#### 3.3.1. Impulsive Choice

We conducted three separate hierarchical multiple regressions with different indices of impulsive choice as dependent variables: (1) decision making under ambiguity (*IGT Block 3, 4, 5*); (2) decision making under risk (*CGT Quality of decision-making*); (3) delay discounting (*MCQ Overall k*).

***Iowa Gambling Task* (IGT Block 3, 4, 5).** The model in Step 1 was significant [F_(3,103)_ = 4.54, *p* = 0.005; R^2^_adjusted_ = 0.091], explaining 9.1% of the variance in performance. The only significant predictor was *fluid intelligence* (ß = 0.322, *p* = 0.001), with higher IQ being associated with better decision making under ambiguity. Step 2 [F_(5,101)_ = 2.75, *p* = 0.023; R^2^_adjusted_ = 0.076] was also significant, but the change in R^2^ did not reach the required level of significance. Step 3 [F_(6,100)_ = 3.04, *p* = 0.009; R^2^_adjusted_ = 0.103] was significant, indicating a significant change in R^2^. *Fluid intelligence* (ß = 0.325, *p* = 0.001) and *length of abstinence* (ß = 0.205, *p* = 0.047) were both significant predictors, with longer periods of abstinence being associated with improved performance on the IGT task. The overall model explained a total of 10.3% of the variance in the quality of decision making under ambiguity (see Table 4).

***Cambridge Gambling Task*** (**CGT Quality of Decision-Making**)**.** Neither Step 1 [F_(3,105)_ = 1.07, *p* = 0.366; R^2^_adjusted_ = 0.002] nor Step 2 [F_(5,103)_ = 2.16, *p* = 0.068; R^2^_adjusted_ = 0.051] were significant. The model in Step 3 was significant [F_(6,104)_ = 3.29, *p* = 0.005; R^2^_adjusted_ = 0.113]. The inclusion of length of abstinence resulted in a significant change in R^2^. Longer periods of abstinence (ß = 0.285, *p* = 0.005) and more opiate dependence symptoms (ß = 0.256, *p* = 0.008) were associated with improved decision-making quality. The overall model explained a total of 11.3% of the variance in the *CGT Quality of decision-making* index (see Table 4).

***Monetary Choice Questionnaire*** (**MCQ Overall *k***)**.** The models in Step 1 [F_(3,98)_ = 2.27, *p* = 0.085; R^2^_adjusted_ = 0.036], Step 2 [F_(5,96)_ = 1.45, *p* = 0.214; R^2^_adjusted_ = 0.022], and Step 3 [F_(6,95)_ = 1.20, *p* = 0.305; R^2^_adjusted_ = 0.012] were not significant. The only significant predictor was *fluid intelligence* (ß = −0.257, *p* = 0.013), with higher IQ being associated with reduced delay discounting (see Table 4).

#### 3.3.2. Impulsive Action

We conducted two hierarchical multiple regressions with different indices of impulsive action as dependent variables: (1) action inhibition (i.e., automatic inhibition) (*GNG False Alarms*); (2) action cancellation (i.e., controlled inhibition) (*SST 150 ms Inhibition*).

***Go/No-Go Task*** (**GNG False Alarms**)**.** Step 1 was significant [F_(3,100)_ = 5.64, *p* = 0.001; R^2^_adjusted_ = 0.119], explaining 11.9% of the variance in the *GNG False Alarms*. The only significant predictor was *fluid intelligence* (ß = −0.374, *p* < 0.001), with higher IQ being associated with increased response inhibition. The models in Step 2 [F_(5,98)_ = 3.60, *p* = 0.005; R^2^_adjusted_ = 0.112] and Step 3 [F_(6,97)_ = 2.98, *p* = 0.010; R^2^_adjusted_ = 0.103] were also significant, but the change in R^2^ did not reach the required level of significance (*p* > 0.05). Fluid intelligence remained the only significant predictor in the model (see Table 5).

***Stop Signal Task*** (**SST 150 ms Inhibition**)**.** Models in Step 1 [F_(3,104)_ = 1.23, *p* = 0.301; R^2^_adjusted_ = 0.007], Step 2 [F_(5,102)_ = 0.83, *p* = 0.533; R^2^_adjusted_ = −0.008], and Step 3 [F_(6,101)_ = 0.79, *p* = 0.581; R^2^_adjusted_ = -.012] were not significant. There were no significant predictors of the ability to cancel an already initiated motor response (see Table 5).

## 4. Discussion

The aims of the current study were to examine the effects of length of abstinence on neurocognitive domains of impulsivity (i.e., impulsive choice and impulsive action) among mono-dependent opiate users at different lengths of abstinence. Our findings suggest that length of abstinence had effects on decision making under risk and ambiguity but was unrelated to delay discounting and response inhibition among abstinent heroin users.

### 4.1. Effects of Length of Abstinence on Impulsive Choice

Within the domain of *impulsive choice*, increased length of abstinence was associated with improved decision making both under ambiguous and explicit reward contingencies, whereas delay discounting was unrelated to length of abstinence. These results reveal that the tendency to discount delayed rewards may remain relatively stable even in periods of protracted abstinence. This suggests that abstinent opiate users may tend to neglect future delayed rewards and adapt a long-term disadvantageous decision-making strategy, associated with increased sensitivity to immediate reinforcements. These findings are consistent with previous studies that have reported a similar trend in samples of heroin dependent individuals in early and protracted abstinence [50,55]. However, our results cannot be interpreted in terms of potential recovery or lack of recovery of function due to the limitations of our cross-sectional design. Future studies would benefit from longitudinal designs examining the trajectory of potential changes in delay discounting with increasing periods of abstinence. Such studies may have important practical implications for the treatment and rehabilitation of opiate dependent individuals, due to the high predictive validity of delay discounting in relation to risk of relapse [78] and other significant treatment outcomes [79,80].

Our data also suggest that opiate users in early remission [<12 months of abstinence] are characterized by disadvantageous decision making under both risk (CGT) and ambiguity (IGT) compared to control participants. These results are consistent with previous findings and support the assumption that decision making remains significantly impaired within the first year of abstinence following chronic opiate use [50,61,62,67]. Contrary to expectations, we did not find any group differences in decision making under risk and ambiguity between the control group and the group of opiate users in sustained remission [>12 months of abstinence]. Results from the limited number of previous studies addressing the stage of protracted abstinence [>12 months] from opiates have produced inconsistent results, with some suggesting that disadvantageous decision making is robust and persists even after prolonged abstinence from opiate use [50,58], whereas others report intact decision making in heroin users in protracted abstinence [67].

Regression analyses revealed that length of abstinence affected performance on decision-making tasks, though these effects were relatively minor. Therefore, it appears that length of abstinence is not the main factor affecting decision-making quality among opiate users. Given that performance on impulsive choice tasks is determined by a number of additional factors and involves multiple processes and mechanisms such as fluid intelligence, inhibitory control, working memory, emotional processing [81], future research needs to assess the relative involvement of these mechanisms in the quality of decision making and delay discounting in opiate dependent individuals.

In summary, current results suggest that there is a weak relationship between length of abstinence and impulsive choice in opiate users. This is in line with findings from neuroimaging studies that report persistent impairments in the structure and function of the orbitofrontal cortex in SDIs, significantly implicated in impulsive choice tasks [82,83,84]. These studies suggest that prolonged exposure to the toxicological effects of various psychoactive substances (including opiates) leads to alterations in brain functioning which may underlie the maladaptive behaviors and disadvantageous decisions that characterize the daily lives of people suffering from SUDs. However, impaired decision making and increased delay discounting can also be regarded as risk factors that precede the onset of SUDs and may explain the tendency of substance users to continue drug use despite negative long-term consequences. In this context, the lack of performance differences between control participants and opiate users in protracted abstinence in this study may be due not to the effects of length of abstinence but may rather reflect stable premorbid neurocognitive features of individuals who are able to successfully maintain prolonged periods of abstinence. Accordingly, opiate users who are able to abstain from drug use for extended periods of time might be characterized by intact or more adaptive decision making at baseline, which in turn may explain their ability to successfully maintain long-term abstinence. It is important to emphasize that the specifics of our research design combined with the methodological limitations of previous studies hinder the identification of the mechanisms underlying the higher impulsive choice of opiate dependent individuals. Their potential persistent impairments in this domain require longitudinal studies that could adequately address such questions.

### 4.2. Effects of Length of Abstinence on Impulsive Action

Our findings in the *impulsive action* domain indicate that regardless of their length of abstinence, abstinent opiate users were characterized by diminished ability to inhibit a prepotent motor response (i.e., automatic inhibition), whereas the ability to cancel an already initiated motor response (i.e., controlled inhibition) was intact.

The observed performance differences in automatic inhibition between control participants and abstinent opiate users contradict previous findings [45,46,48,66] and may reflect task-specific effects. In contrast to previous studies [45,46,66] that used relatively easy Go/No-Go paradigms, the task in the current study included No-Go conditions of varying levels of difficulty. In support of this hypothesis, the difference between abstinent opiate users and control participants was observed only on the “hard” No-Go trials. These findings suggest that when the task was more cognitively demanding, abstinent opiate users show impaired performance compared to the control group, but when the task was relatively easy, response inhibition appears to be intact. It is important to note that the effects of these differences were moderate, suggesting that length of abstinence is not a key factor affecting performance on tasks measuring automatic inhibition. This was further supported by the regression analyses, which identified fluid intelligence as the only significant predictor of the ability to inhibit a dominant motor response. Therefore, future research needs to investigate the effects of other relevant factors (e.g., emotional processes, attention, memory) that might influence automatic inhibition by examining their interaction effects with length of abstinence from chronic opiate use.

On the other hand, the lack of group differences on the SST task suggests that controlled inhibition is not significantly impaired in former opiate users. These results are consistent with the findings of Ahn and Vassileva [48], who reported intact controlled inhibition among opiate users in protracted abstinence. However, it is important to note that, to our knowledge, the majority of previous studies examining impulsive action in opiate users have only used Go/No-Go paradigms to capture response inhibition. Therefore, our knowledge about the quality of controlled inhibition in opiate users is limited and requires further investigation in different samples of opiate dependent individuals. Nevertheless, the current results, combined with the findings of Ahn and Vassileva [48], suggest that controlled inhibition associated with the ability to cancel an already initiated motor response is relatively preserved in abstinent opiate users.

The observed variations in performance on distinct impulsive action tasks may be due to different residual effects of opiates on the neural circuits mediating performance on Stop Signal vs. Go/No-Go paradigms. Alternatively, both types of response inhibition may be significantly impaired among opiate users, but the ability to cancel an already initiated response may recover earlier in the course of abstinence. In this context, the relatively long periods of abstinence maintained by our opiate dependent participants may explain the lack of group differences on the controlled inhibition task (SST). To address these gaps in the literature, future research could additionally examine different types of response inhibition in current and abstinent opiate users. In addition, using longitudinal designs and neuroimaging approaches could significantly improve our knowledge about different types of response inhibition impairments in opiate addiction and their recovery with abstinence.

In summary, our main findings indicate that abstinent opiate users are characterized by long-lasting impairments in automatic inhibition involving action inhibition, whereas they manifest intact controlled inhibition processes that engage action cancellation. In addition, length of abstinence may have no significant effects on response inhibition, with poorer automatic inhibition persisting into protracted abstinence.

### 4.3. Limitations and Future Directions

The current study has several limitations that need to be considered. First, we used only neurobehavioral methods to examine neurocognitive impulsivity. Future studies should include additional levels of analysis by combining behavioral assessment with genetic and neuroimaging approaches, which may increase our understanding of the heterogeneity of addictions and facilitate the development of targeted interventions that are sensitive to the individual genetic, neurobiological, and neurobehavioral vulnerabilities [2]. Second, our study did not control for additional cognitive and affective processes that may influence or mediate the deficits in impulsive choice and impulsive action among abstinent opiate users. Future studies should include the assessment of additional cognitive functions such as working memory, attention, and emotional processes known to affect performance on impulsive choice and impulsive action tasks. Third, our study used a cross-sectional design, which limits our knowledge regarding the effects of length of abstinence on distinct neurocognitive dimensions of impulsivity. Current results may not reflect potential changes associated with the recovery of decision making and response inhibition in the course of abstinence but may rather represent specific premorbid characteristics of individuals who are able to successfully maintain prolonged periods of abstinence. Therefore, future research should examine the dimensions of neurocognitive impulsivity in former opiate users longitudinally to further understand the trajectory of recovery of neurocognitive function with abstinence. Fourth, the participant groups in the current study were not well matched in terms of years of education and fluid intelligence, which resulted in significant group differences in these variables. Although our samples are representative and reflect the real-world tendency of lower education and lower fluid intelligence noted among opiate users, future studies may control these variables to prevent inconsistent findings. Fifth, our study did not include comprehensive evaluation of co-occurring mental health disorders that are highly comorbid with opiate dependence, such as mood disorders, anxiety disorders and personality disorders. Future studies could evaluate more thoroughly the confounding effects of other mental health disorders on neurocognitive impulsivity among opiate users. Another limitation of the current study is the lack of control for the effects of different treatment interventions on neurocognitive aspects of impulsivity. Given that 32.7% of our participants were enrolled in nonpharmacological treatment at the time of testing, the psychosocial interventions included in these programs may have exerted some effects on their neurocognitive functioning. Therefore, future studies should additionally examine the effects of various pharmacological and nonpharmacological interventions on distinct neurocognitive impulsivity domains at different lengths of abstinence. Finally, our opiate dependent group in sustained remission was highly heterogeneous in terms of length of abstinence, with abstinence periods ranging between 12 months and 9 years. Future studies should collect data in relatively more homogeneous groups of abstinent opiate users that reflect different stages of the recovery process (e.g., 1–2 years of abstinence, 2–3 years of abstinence, etc.). This research strategy will allow for the development of a more consistent picture of the profiles of strengths and weaknesses in neurocognitive impulsivity in opiate users, depending on the specific stage of protracted abstinence.

## 5. Conclusions

In summary, our results suggest that some residual deficits in neurocognitive functioning are long-lasting and not significantly affected by length of abstinence, indicating that they may exert their negative effects even after years of successful recovery from chronic opiate use. In line with prior research, current findings underscore the increased need for the development of novel personalized cognitive rehabilitation programs for former opiate users in protracted abstinence. The development of tailored modular interventions addressing deficits in cognitive functioning may have broad practical implications for the rehabilitation of opiate addiction and may help address some of the limitations of traditional therapeutic approaches.

## Figures and Tables

**Table 1 ijerph-20-01236-t001:** Descriptive statistics and group differences in demographic characteristics and opiate use variables.

	Control Group (1)	Early Full Remission (2)	Sustained Full Remission (3)	*p*	Contrast
**N**	68	45	68	-	-
**Age**	30.59 (5.79)	31.31 (5.62)	32.06 (5.26)	0.159	-
**Gender** **(N (% females))**	17 (25%)	9 (20%)	13 (19.1%)	0.677	-
**Years of** **education**	15.07 (2.50)	12.40 (2.12)	13.46 (2.69)	**<0.001**	1 > 2, 3
**Fluid intelligence (IQ)**	111.46 (10.73)	104.47 (11.93)	103.61 (12.53)	**<0.001**	1 > 2, 3
**Severity of opiate dependence (Number of symptoms)**	-	6.24 (1.00)	6.01 (1.02)	0.161	-
**Duration of** **opiate** **dependence (years)**	-	8.83 (4.08)	7.64 (4.11)	0.094	-
**Length of** **abstinence (months)**	-	6.56 (3.76)	53.60 (30.74)	**<0.001**	2 < 3

Note: Results are presented as means (SD). Values in bold are significant.

**Table 2 ijerph-20-01236-t002:** Descriptive statistics and group differences on indices of impulsive choice.

	Control Group (1)	Early Full Remission (2)	Sustained Full Remission (3)	*p*	Contrast
**N**	66	44	66	-	
**CGT Quality of decision making**	0.91 (0.10)	0.83 (0.16)	0.87 (0.14)	**0.005**	1 > 2
**N**	63	39	64		
**MCQ Overall k**	0.048 (0.073)	0.086 (0.081)	0.075 (0.087)	**0.001**	1 < 2, 3
**N**	67	43	65		
**IGT Net score**	10.04 (26.97)	−4.74 (19.65)	1.37 (28.95)	**0.013**	1 > 2

Note: CGT Quality of decision making = Cambridge Gambling Task Quality of Decision-Making index; MCQ Overall k = Monetary Choice Questionnaire Overall k index; IGT Net score = Iowa Gambling Task Net Score. Results are presented as means (SD). Values in bold are significant.

**Table 3 ijerph-20-01236-t003:** Descriptive statistics and group differences on indices of impulsive action.

	Control Group (1)	Early Full Remission (2)	Sustained Full Remission (3)	*p*	Contrast
**N**	65	42	65	-	
**GNG False alarms**	12.82 (7.54)	17.98 (13.19)	15.71 (7.63)	**0.015**	1 < 2, 3
**GNG Hard No-Go False Alarms**	2.83 (3.22)	5.95 (11.79)	3.57 (3.16)	**0.025**	1 < 2, 3
**GNG Easy No-Go False Alarms**	9.98 (5.07)	12.02 (6.14)	12.14 (5.24)	0.098	-
**N**	67	44	67	-	
**SST 150 ms Inhibition**	72.76 (23.18)	71.93 (19.51)	74.40 (18.12)	0.792	-

Note: GNG False Alarms **=** Go/No-Go False Alarms index; GNG Hard No-Go False Alarms = Go/No-Go False Alarms index in the Hard-No-Go condition; GNG Easy No-Go False Alarms = Go/No-Go False Alarms index in the Easy-No-Go condition; SST 150 ms Inhibition = Stop Signal Task 150 ms Inhibition index. Results are presented as means (SD). Values in bold are significant.

**Table 4 ijerph-20-01236-t004:** Length of abstinence as predictor of IGT Net Score, CGT Quality of Decision-Making, and MCQ Overall k.

	IGT Net Score				CGT Quality of Decision-Making				MCQ Overall k			
	B	SE B	β	∆R^2^	B	SE B	β	∆R^2^	B	SE B	β	∆R^2^
**Step 1**				0.12 **				0.03				0.05
Age	−0.34	0.38	−0.09		−0.01	0.00	−0.11		0.00	0.01	0.03	
Biological sex	−4.69	4.84	−0.09		0.04	0.04	0.12		0.06	0.14	0.04	
Fluid intelligence (IQ)	0.54	0.16	0.32 **		0.00	0.00	0.01		−0.01	0.01	−0.25 *	
**Step 2**				0.00				0.07 *				0.01
Age	−0.46	0.44	−0.12		0.00	0.00	0.01		0.01	0.01	0.07	
Biological sex	−4.85	4.91	−0.10		0.05	0.04	0.14		0.06	0.14	0.04	
Fluid intelligence (IQ)	0.55	0.16	0.33 **		<0.001	0.00	0.00		−0.01	0.01	−0.26 *	
Duration of HD	0.00	0.00	0.05		<0.001	0.00	−0.14		<0.001	0.00	−0.07	
Severity of HD	−0.81	1.94	−0.04		0.04	0.01	0.25 *		0.02	0.06	0.04	
**Step 3**				0.03 **				0.07 **				<0.001
Age	−0.80	0.46	−0.20		0.00	0.00	−0.10		0.01	0.01	0.08	
Biological sex	−7.43	5.01	−0.15		0.03	0.04	0.07		0.07	0.15	0.05	
Fluid intelligence (IQ)	0.57	0.16	0.34		0.00	0.00	0.02		−0.01	0.01	−0.26 *	
Duration of HD	0.00	0.00	0.10		<0.001	0.00	−0.07		<0.001	0.00	−0.08	
Severity of HD	−0.74	1.91	−0.04		0.04	0.01	0.26 **		0.02	0.06	0.04	
Length of abstinence	0.00	0.00	0.21 *		0.00	0.00	0.29 **		<0.001	0.00	−0.02	

Note. IGT Net score = Iowa Gam-bling Task Net Score; CGT Quality of decision-making = Cambridge Gambling Task Quality of Decision-Making index; MCQ Overall k = Monetary Choice Questionnaire Overall k index; Biological sex = Male (1), Female (2); HD = heroin dependence; * *p* < 0.05; ** *p* < 0.01.

**Table 5 ijerph-20-01236-t005:** Length of abstinence as predictor of GNG False Alarms and SST 150 ms Inhibition.

	GNG False Alarms				SST 150 ms Inhibition			
	B	SE B	β	∆R^2^	B	SE B	β	∆R^2^
**Step 1**				0.15 **				0.03
Age	0.03	0.15	0.02		0.28	0.31	0.09	
Biological sex	2.26	1.89	0.11		−5.53	4.13	−0.13	
Fluid intelligence (IQ)	−0.24	0.06	−0.37 **		−0.06	0.14	−0.05	
**Step 2**				0.01				0.01
Age	−0.05	0.17	−0.03		0.39	0.37	0.12	
Biological sex	2.11	1.91	0.11		−5.67	4.19	−0.14	
Fluid intelligence (IQ)	−0.24	0.06	−0.37 **		−0.07	0.14	−0.05	
Duration of HD	0.00	0.00	0.07		<0.001	0.00	−0.08	
Severity of HD	−0.73	0.74	−0.10		−0.01	1.65	<0.001	
**Step 3**				0.00				0.01
Age	−0.03	0.19	−0.02		0.50	0.40	0.16	
Biological sex	2.23	2.00	0.11		−4.80	4.34	−0.12	
Fluid intelligence (IQ)	−0.24	0.06	−0.37 **		−0.08	0.14	−0.06	
Duration of HD	0.00	0.00	0.07		<0.001	0.00	−0.10	
Severity of HD	−0.73	0.75	−0.10		−0.08	1.66	<0.001	
Length of abstinence	0.00	0.00	−0.02		<0.001	0.00	−0.08	

Note. GNG False Alarms = Go/No-Go False Alarms index; SST 150 ms Inhibition = Stop Signal Task 150 ms Inhibition index; Biological sex = Male (1), Female (2); HD = heroin dependence; ** *p* < 0.01.

## Data Availability

Deidentified data are available upon request to qualified investigators by contacting the first author.

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
