# Peer review of "Neurocognitive Impulsivity in Opiate Users at Different Lengths of Abstinence"

_ijerph, 2023, doi:10.3390/ijerph20021236_

Round 1

Reviewer 1 Report

Dear Editor

Please convey my evaluation of the article titled "Neurocognitive Impulsivity in Opiate Users at Different Stages of Abstinence" to the authors.

In the article, the authors report that the data were collected during a large international survey. Therefore, it would be appropriate to cite these previous study or studies under the sub-title of "Participants" in the title "Materials and Methods" appropriately. In addition, since it was reported that DSM-IV was used in the study, the dates and places of the study should be reported in order to understand the time of the study. The second paragraph of the "Participants" section contains some data on the results of the study, it may be appropriate to transfer some of it to the "Results" section.

As understood from the method section, whether the participants had an impulse control disorder or not was not evaluated as an inclusion or exclusion criterion. In addition, the presence of a possible major depression in the participants or another severe co-morbid psychiatric disorder may adversely affect the evaluation of the data. These situations are not given as a criterion for participation in the study and should be reported as limitations. .

Drug abuse and substance dependence were two distinct diagnoses for substance use disorder in the DSM-IV. These two disorders were consolidated into one in the DSM-5 to form a single diagnostic category for substance use disorder. In order to avoid confusion for the reader, it would be appropriate for the authors to add a sentence or two to clarify this issue.

Minor suggestions

At line 557 “direcRtions” should be corrected.

It would be appropriate to give the long versions of the abbreviations in the tables.

Author Response

We highly appreciate the reviewer’s comments and recommendations and have revised the manuscript accordingly.

First, we included references to previous studies within the larger research project, which contain detailed description of the study protocol and our prior findings (in Procedures subsection). We have now added information about the time period during which the data were collected (in Participants subsection).

Second, we have now transferred some of the participants data to the Results section.

Third, we have now included a brief discussion of one of the main limitations of the study related to the lack of comprehensive evaluation of comorbid mental health disorders (in Limitations and Future Directions section).

Fourth, we have now included a brief description of the DSM-IV and DSM-5 criteria for substance use disorders and their main differences (in Instruments section).

Finally, we included the full forms of the abbreviations of the main neurocognitive tests in Notes under the tables.

Reviewer 2 Report

The authors outline in a consice and understandable manner the relevence of the neuripsychological construct of impulsivity for SUDs as a group and OUD in particular. The selected methodology is adequate for a cross-sectional study, so are the group sizes. A little concern may be the higher (though not statistically significant) mean duration of education of the control group, but this is not surprising given the impairment of functioning (including schooling) associated with OUDs. The results are well articulated and the conclusions are logically derived from them. The limitations of the study resulting from the methodology and design are also emphasized. 

Author Response

We highly appreciate the reviewer’s comments. We have now included a brief section in Limitations and Future Directions which emphasizes the limitations of the current study related to the group differences in education and fluid intelligence.

Reviewer 3 Report

Elena Psederska and Jasmin Vassileva conducted this study "to examine the effects of length of abstinence on neurocognitive domains of impulsivity (i.e., impulsive choice and impulsive action) among mono-dependent opiate users at different stages of abstinence." The method sounds appropriate, and the manuscript is written well and fluent. There are just a few minor issues that would be better to be addressed:

1. The method of acute detoxification in abstinent groups would be better to mention.

2. since the study is about the effect of the length of abstinence, it is important to mention if there are any medical interventions (such as naltrexone) or non-medical interventions (such as NA or psychological intervention). This is not only missing in the analyses, but also it has not been discussed in the discussion.

3. Statistical analysis in the methodology section is not well described. Although, there are mixture of statistical analysis and results in the results section, it would be better if methodology can be separated from the result. 

Author Response

We highly appreciate the reviewer’s comments and recommendations and have revised the manuscript accordingly.

First, we have now provided information about the pharmacological and non-pharmacological treatments in which participants were enrolled at the time of testing. In addition, we included a brief paragraph in the Limitations and Future Directions section, which discusses one of the main limitations of the study related to the lack of control on the effects of treatment interventions on neurocognitive impulsivity among former opiate users.

Second, we have now described all statistical analyses in Data Analytic Plan section and left all results in the Results section.

Unfortunately, we don’t have information about the method of acute detoxification of our participants (most of our participants were in protracted abstinence at the time of testing) which prevented us from providing more details on this topic.